# SpECTra: Sparse Entity-centric Transitions

## Abstract

Learning an agent that interacts with objects is ubiquituous in many RL tasks. In most of them the agent's actions have **sparse** effects : only a small subset of objects in the visual scene will be affected by the action taken. We introduce SPECTRA, a model for learning *slot-structured* transitions from raw visual observations that embodies this sparsity assumption. Our model is composed of a perception module that decomposes the visual scene into a set of latent objects representations (i.e. **slot-structured**) and a transition module that predicts the next latent set slot-wise and in a sparse way. We show that learning a perception module jointly with a sparse slot-structured transition model not only biases the model towards more *entity-centric* perceptual groupings but also enables intrinsic exploration strategy that aims at maximizing the number of objects changed in the agents trajectory.

## 1 Introduction

Recent model-free deep reinforcement learning (DRL) approaches have achieved human-level performance in a wide range of tasks such as games (Mnih et al., 2015). A critical known drawback of these approaches is the vast amount of experience required to achieve good performance. The promise of model-based DRL is to improve sample-efficiency and generalization capacity across tasks. However model-based algorithms pose strong requirements about the models used. They have to make accurate predictions about the future states which can be very hard when dealing with high dimensional inputs such as images. Thus one of the core challenge in model-based DRL is learning accurate and computationally efficient transition models through interacting with the environment. Buesing et al. (2018) developed state-space models techniques to reduce computational complexity by making predictions at a higher level of abstraction, rather than at the level of raw pixel observations. However these methods focused on learning a state-space model that doesn't capture the compositional nature of observations: the visual scene is represented by a single latent vector and thus cannot be expected to generalize well to different objects layouts.

Extensive work in cognitive science (Baillargeon et al., 1985; Spelke, 2013) indeed show that human perception is structured around objects. Object-oriented MDPs (Diuk et al., 2008) show the benefit of using object-oriented representations for structured exploration although the framework as it is presented requires hand-crafted symbolic representations. Bengio (2017) proposed as a prior (the consciousness prior) that the dependency between high-level variables (such as those describing actions, states and their changes) be represented by a *sparse factor graph*, i.e., with few high-level variables at a time interacting closely, and inference performed sequentially using attention mechanisms to select a few relevant variables at each step.

Besides, a recent line of work (Greff et al., 2017; van Steenkiste et al., 2018; Eslami et al., 2016; Kosiorek et al., 2018; Greff et al., 2019; Burgess et al., 2019) has focused on unsupervised ways to decompose a raw visual scene in terms of objects. They rely on a **slot-structured** representation (see Figure 1) of the scene where the latent space is a set of vectors and each vector of the set is supposed to represent an "object" (which we refer to as "entity") of the scene. However, to the best of our knowledge, Watters et al. (2019) is the only work that investigates the usefulness of slot-structured representations for RL. They introduced a method to learn a transition model that is applied to all the slots of their latent scene representation. Extending their work, we go further and posit that slot-wise transformations should be sparse and that the perception module should be learned jointly with the transition model.

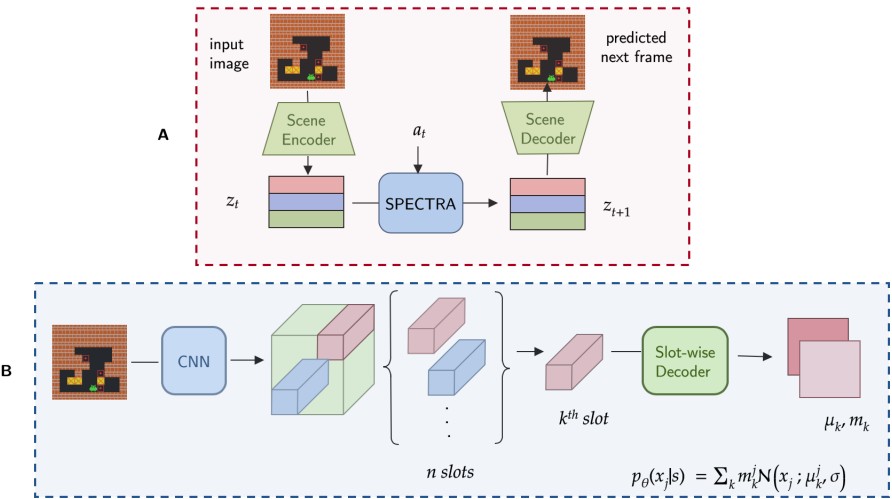

Figure 1: **A**: SPECTRA. Illustration of an entity-centric transition model. **B**: Naive Perception module with a CNN-based encoder and a slot-wise decoder. Hyperparameters description in Appendix A.

We introduce **Sp**arse **E**ntity-**C**entric **Tra**nsitions (**SPECTRA**), an entity-centric action-conditioned transition model that embodies the fact that the agents actions have sparse effects: that means that each action will change only a few slots in the latent set and let the remaining ones unchanged. This is motivated by the physical consideration that agent interventions are localized in time and space. Our contribution is motivated by three advantages:

— Sparse transitions enable transferable model learning. The intuition here is that the sparsity of the transitions will bias the model towards learning primitive transformations (e.g. how pushing a box affects the state of a box being pushed etc) rather than configuration-dependent transformations, the former being more directly transferable to environments with increased combinatorial complexity.

— Sparse transitions enable a perception module (when trained jointly) to be biased towards more meaningful perceptual groupings, thus giving potentially better representations that can be used for downstream tasks, compared to representations learned from static data.

— Sparse transitions enable an exploration strategy that learns to predict actions that will change the state of as many entities as possible in the environment without relying on pixels error loss.

## 2 RELATED WORK

**Unsupervised visual scene decomposition.** Learning good representations of complex visual scenes is a challenging problem for AI models that is far from solved. Recent work (Greff et al., 2017; van Steenkiste et al., 2018; Eslami et al., 2016; Kosiorek et al., 2018; Greff et al., 2019; Burgess et al., 2019) has focused on learning models that discover objects in the visual scene. Greff et al. (2019) further advocates for the importance of learning to segment and represent objects jointly. Like us they approach the problem from a spatial mixture perspective. van Steenkiste et al. (2018) and Kosiorek et al. (2018) build upon Greff et al. (2017) and Eslami et al. (2016) respectively by incorporating next-step prediction as part of the training objective in order to guide the network to learn about essential properties of objects. As specified in van Steenkiste et al. (2019) we also believe that objects are task-dependent and that learning a slot-based representations along with sparse transitions bias the perception module towards **entity-centric** perceptual groupings and that those structured representations could be better suited for RL downstream tasks.

**Slot-based representation for RL.** Recent advances in deep reinforcement learning are in part driven by a capacity to learn good representations that can be used by an agent to update its policy. Zambaldi et al. (2018) showed the importance of having structured representations and computation when it comes to tasks that explicitly targets relational reasoning. Watters et al. (2019) also show the importance of learning representations of the world in terms of objects in a simple model-based setting. Zambaldi et al. (2018) focuses on task-dependent structured computation. They use a self-attention mechanism (Vaswani et al., 2017) to model an actor-critic based agent where vectors in the set are supposed to represent entities in the current observation. Like Watters et al. (2019) we take a model-based approach: our aim is to learn task-independent slot-based representations that can be further used in downstream tasks. We leave the RL part for future work and focus on how learning those representations jointly with a sparse transition model may help learn a better transition model.

# 3   SPECTRA

Our model is composed of two main components: a **perception module** and a **transition module** (section 3.1). The way we formulated the transition implicitly defines an **exploration policy** (section 3.3) that aims at changing the states of as many entities as possible.

**Choice of Environment.** Here we are interested in environments containing entities an agent can interact with and where actions only affect a *few* of them. **Sokoban** is thus a good testbed for our model. It consists of a difficult puzzle domain requiring an agent to push a set of boxes onto goal locations. Irreversible wrong moves can make the puzzle unsolvable. Each room is composed of walls, boxes, targets, floor and the agent avatar. The agent can take 9 different actions (no-op, 4 types of push and 4 types of move).

**Fully Observed vs Learned Entities.** The whole point is to work with slot-based representations learned from a raw pixels input. There is no guarantee that those learned slots will effectively correspond to entities in the image. We thus distinguish two versions of the environment (that correspond to two different levels of abstraction):

- **Fully observed entities**: the input is structured. Each entity corresponds to a spatial location in the grid. Entities are thus represented by their one-hot label and indexed by their x-y coordinate. This will be referred to as the *fully observed setting*. There is no need for a perception module in this setting.

- **Raw pixels input**: the input is unstructured. We need to infer the latent entities representations. This will be referred to as the *latent setting*.

## 3.1   MODEL OVERVIEW

The idea is to learn an action-conditioned model of the world where at each time step the following take place:

- **Pairwise Interactions**: Each slot in the set gathers relevant information about the slots conditioned on the action taken
- **Active entity selection** : Select slots that will be modified by the action taken
- **Update**: Update the selected slots and let the other ones remain unchanged.

Ideally, slots would correspond to unsupervisedly learned entity-centric representations of a raw visual input like it is done by Burgess et al. (2019); Greff et al. (2019). We show that learning such perception modules jointly with the sparse transition biases the perceptual groupings to be entity-centric.

**Perception module.** The perception module is composed of an encoder $f_{enc}$ and a decoder $f_{dec}$. The encoder maps the input image $\mathbf{x}$ to a set of $\mathbf{K}$ latent entities such that at time-step $t$ we have $f_{enc}(\mathbf{x}^t) = \mathbf{s}^t \in \mathbb{R}^{K \times p}$. It thus outputs a **slot-based** representation of the scene where each slot is represented in the same way and is supposed to capture properties of one *entity* of the scene. Like (Burgess et al., 2019; Greff et al., 2019) we model the input image $\mathbf{x}^t$ with a spatial Gaussian

Mixture Model. Each slot $s_k^t$ is decoded by the same decoder $f_{dec}$ into a pixel-wise mean $\mu_{ik}$ and a pixel-wise assignment $m_{ik}^t$ (non-negative and summing to 1 over $k$). Assuming that the pixels $i$ are independent conditioned on $\mathbf{s}^t$, the conditional likelihood thus becomes:

$$p_\theta(\mathbf{x}^t|\mathbf{s}^t) = \prod_{i=1}^{D} \sum_k m_{ik}^t \mathcal{N}(\mathbf{x_i^t}; \mu_{ik}^t, \sigma^2) \text{ with } \mu_{ik}^t, m_{ik}^t = f_{dec}(s_k^t)_i.$$

As our main goal is to investigate how sparse transitions bias the groupings of entities, in our experiments we use a very simple perception module represented in Figure 1. We leave it for future work to incorporate more sophisticated perception modules.

**Pairwise interactions.** In order to estimate the transition dynamics, we want to select relevant entities (represented at time $t$ by the set $\mathbf{s}^t \in \mathbb{R}^{K \times p}$) that will be affected by the action taken, so we model the fact that each entity needs to gather useful information from entities interacting with the agent ( i.e. is the agent close ? is the agent blocked by a wall or a box ? etc..). To that end we propose to use a self-attention mechanism (Vaswani et al., 2017). From the $k$-th entity representation $s_k^t$ at time $t$, we extract a row-vector *key* $K_k^t$, a row-vector *query* $Q_k^t$ and a row-vector *value* $V_k^t$ conditioned on the action taken such that (aggregating the rows into corresponding matrices and ignoring the temporal indices):

$$\tilde{\mathbf{s}} = softmax(\frac{KQ^T}{\sqrt{d}})V$$

where the softmax is applied separately on each row. In practice we concatenate the results of several attention heads to use it as input to the entity selection phase.

**Entity selection.** Once the entities are informed w.r.t. possible pairwise interactions the model needs to select which of these entities will be affected by the action taken $a^t$. Selection of the entities are regulated by a selection gate (Hochreiter & Schmidhuber, 1997; Cho et al., 2014) computed slot-wise as:

$$f_k^t = \sigma(MLP([\tilde{s}_k^t; a^t])) \tag{1}$$

where $f_k^t$ can be interpreted as the probability for an entity to be selected.

**Update.** Finally, each **selected** entity is updated conditioned on its state $s_k^t$ at time-step $t$ and the action taken $a^t$. We thus simply have:

$$s_k^{t+1} = f_k^t f_\theta([s_k^t, a^t]) + (1 - f_k^t)s_k^t$$

$f_\theta$ is a learned action-conditioned transformation that is applied slot-wise. We posit that enforcing the transitions to be slot-wise and implicitly sparse will bias the model towards learning more primitive transformations. We verify this assumption in next subsection in the simpler case where the entities are fully observed (and not inferred with a perception module).

## 4 EXPERIMENTS

In this work we demonstrate three advantages of entity-centric representations learned by SPEC-TRA:

- Implicitly imposing the transitions to be sparse will enable us to learn transition models that will transfer better to environments with increased combinatorial complexity. Section 4.1.
- Learning slot-based representations jointly with a sparse transition model will bias the perceptual groupings to be *entity-centric*. Section 4.2.
- Finally we investigate the usefulness of the implicit exploration scheme induced by SPEC-TRA when learning the model jointly. Section 4.3.

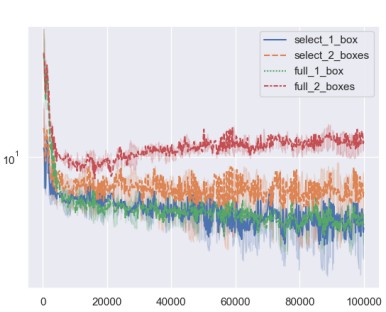 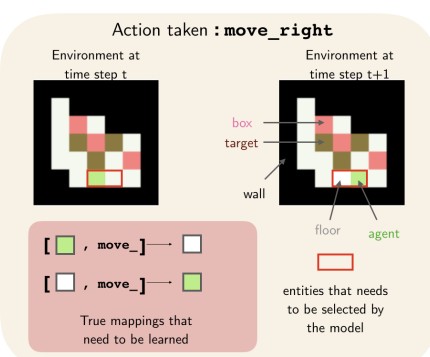

Figure 2: **left**: Full and sparse settings are trained on environment containing one box and evaluated out-of-distribution on two boxes. We plotted the validation losses of both settings during training. The full connectivity architecture is unable to achieve out-of-distribution generalization to an environment with two boxes. **right**: Illustration of what the model has to learn in the fully observed setting: to be correct the model needs to map any concatenation of [agent,move] to a vacated position = floor and to select only the right entities to be changed. The learned mappings are general rules that are directly transferable to settings with more boxes.

## 4.1 LEARNED PRIMITIVE TRANSFORMATIONS

In this section we show that sparse selection in the transitions yields learned slot-wise transformations that are transferable to out-of-distribution settings with increased combinatorial complexity. We restrict ourselves to the **fully observed setting**. Like (Zambaldi et al., 2018) the entities correspond to a spatial location in the $7 \times 7$ grid. Each entity $s_k$ is thus described in terms of its label to which we append its x-y coordinate. The results in Figure 2 are intuitive; to learn the right transitions with our formulation, the model is forced to:

– select only the relevant entities to be updated.
– learn the right primitive transformation (e.g. if the agent slot is selected to be modified by any of the *move* actions, then its position is vacated, so the model should map any concatenation of [*agent*, *move*] to the *floor* label etc...). See Figure 2, right.

Here entity representations are not learned and thus correspond to their labels. We thus train the model with a simple cross-entropy loss. We are interested in comparing two settings:

– **Sparse** setting: the transformation is still done slot-wise to selected entities only. Each slot contains the label and x-y coordinate of the entity only. The transformation is applied to a concatenation of the entity label and the action [*label*,*action*].
– **Full** setting: the transformation is still done slot-wise but this time each slot in $\tilde{s}_t$ potentially contains information about all the other slots in the set. The transformation is applied to a concatenation of the entity representation $\tilde{s}_k^t$ and the action [$tildes_k^t$,*action*]. Thus we hypothesize that the transformation module will learn *configuration-dependent* rules (e.g. if an agent is close to a box and a wall, and 3 steps ahead there is a target to be reached, and it takes a move action to do so) that will not be easily transferable to environments with increased complexity and a wider variety of contexts.

Both settings are illustrated in Figure 7 of the Appendix. In Figure 2 we reported the evolution of training and evaluation losses of both the full and the sparse settings when the models are trained in a 7x7 environment with one box and evaluated in a 7x7 environment with two boxes.

## 4.2 STRUCTURED REPRESENTATION LEARNING

In this section we demonstrate how learning a perception module along with sparse transitions will bias this module towards learning **entity-centric** perceptual groupings of the raw pixel input. In

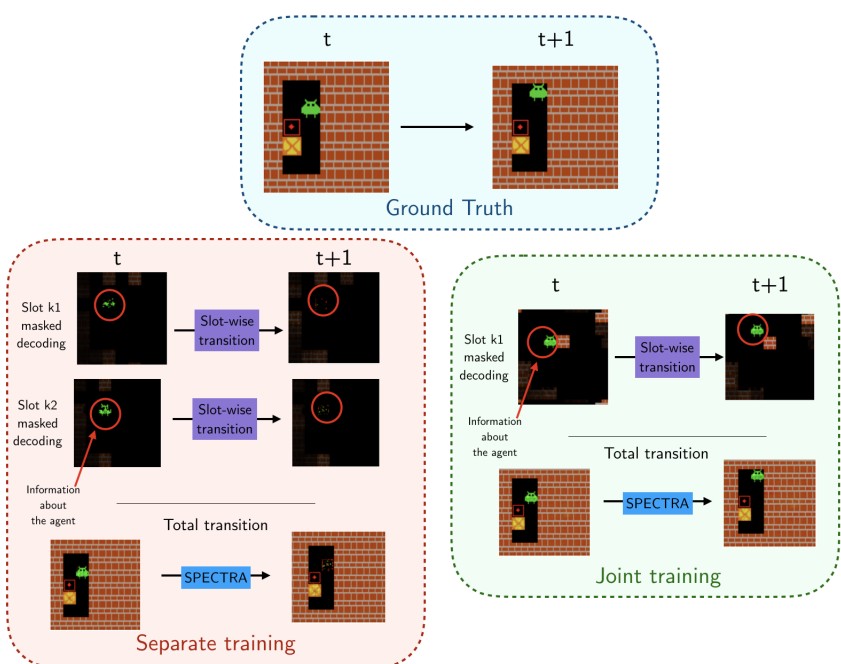

Figure 3: Comparison of slot-wise masked decodings when the perception module is trained separately or jointly with the sparse transitions. We show the reconstruction associated with the slots that contain information about the agent. When the perception module is trained jointly, slots in the learned latent set are biased to be entity-centric (here agent-centric).

order to verify this intuition we compare in Figure 3 the reconstructions from the perception module when it is trained *separately* vs *jointly* with the sparse transition module. In this experiment the input is not structured anymore but just a raw 112x112x3 pixel image. We used a simple perception module as described in Figure 1.

We thus distinguish two losses, a reconstruction loss

$$\mathcal{L}_{percep} = \sum_{i=1}^{D} \log \sum_{k} m_{ik}^t \mathcal{N}(\mathbf{x_i^t}; \mu_{ik}^t, \sigma^2)$$

and a transition loss

$$\mathcal{L}_{trans} = \sum_{i=1}^{D} \log \sum_{k} \hat{m}_{ik}^{t+1} \mathcal{N}(\mathbf{x_i^{t+1}}; \hat{\mu}_{ik}^{t+1}, \sigma^2)$$

with $\mu_{ik}^t, m_{ik}^t = f_{dec}(s_k^t)_i$, $s^t = f_{enc}(\mathbf{x}^t)$, $\hat{\mu}_{ik}^{t+1}, \hat{m}_{ik}^{t+1} = f_{dec}(\hat{s}_k^{t+1})_i$, and $\hat{s}_k^{t+1} = f_{trans}(s_k^t)$ is the future state predicted by the transition function.

$f_{dec}, f_{enc}$ and $f_{trans}$ are respectively the decoder, the encoder and the transition modules. For the joint training (resp. separate training) setting, gradients from $\mathcal{L}_{trans}$ are back-propagated through parameters of $f_{enc}$ and $f_{trans}$ (resp. $f_{trans}$ only). In both settings, gradients from $\mathcal{L}_{percep}$ are back-propagated through parameters of $f_{enc}$ and $f_{dec}$.

In Figure 3 we put particular attention on the masked reconstructions from slots containing *visual information* about the agent. We can directly notice that the perceptual groupings done by the encoder, when it is trained jointly with the transition module, are **agent-centric**: the information about the agent is contained in one slot only (whereas it is often contained in several slots in the separate training settings). Moreover, in Figure 4 we see the joint training setting leading to a better transition model: we hypothesize that the transformations are easier to learn specifically because they have to focus on the effects of the actions taken on entities, i.e., involving a few strongly dependent variables at a time rather than more global but more specific configurations involving all the variables in the state, as suggested by Bengio (2017).

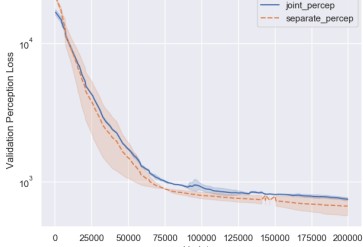 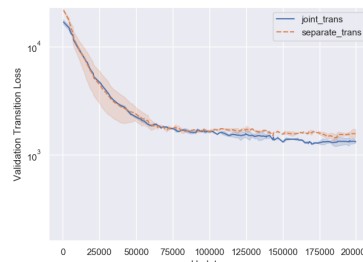

Figure 4: Loss vs training updates, with training is done in pixel space, transitions are sampled randomly and results are averaged over 3 runs. **left**: Validation perception loss $\mathcal{L}_{percep}$ of joint and separate training **right**: Validation transition loss $\mathcal{L}_{trans}$ of joint and separate training. Separate training is better in terms of perception loss but joint training gives a better transition model. We posit that this is because the slots are biased to be **entity-centric** and transformations involving only relevant entities are easier to learn.

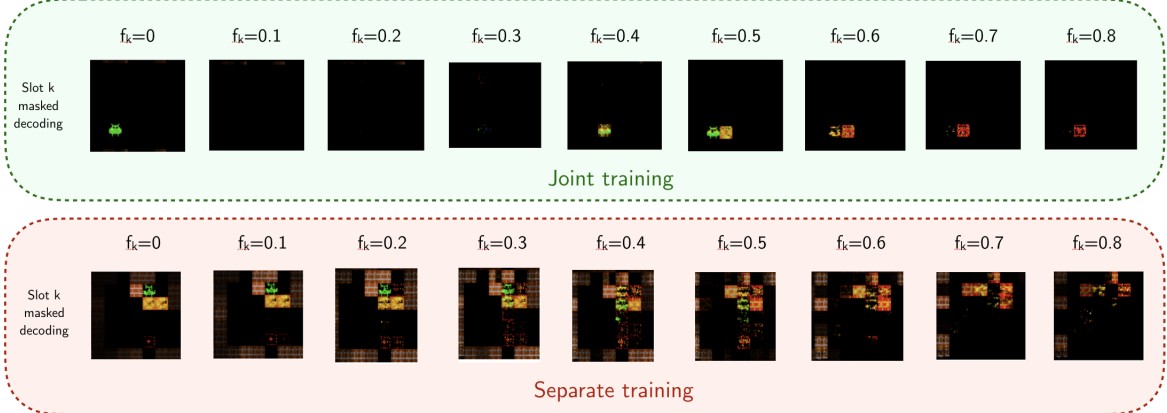

Figure 5: Reconstructions of a slot for different levels of its activation gate (Eq. 1). Transformations learned when the perception module is trained jointly vs separately from the sparse transitions. When the training is done end-to-end, transformations seem to be more interpretable. We posit that this is because the slots are biased to be **entity-centric** as shown in Figure 3. It is interesting to notice that enough changes appear for an activation gate value of $f_k = 0.1$. We may want to explore an explicit sparsity constraint for the selection mechanism.

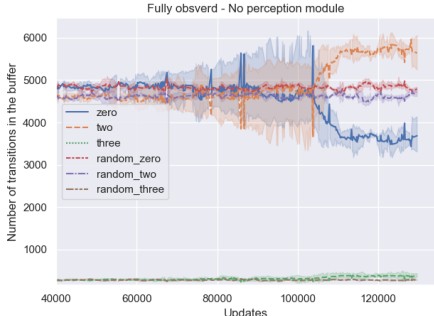 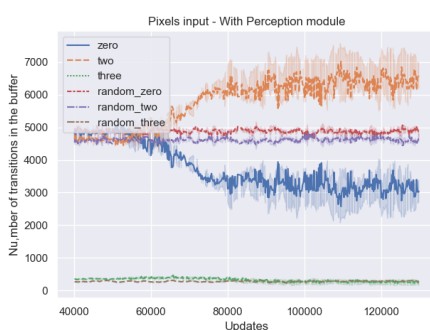

Figure 6: Comparison is done against randomly sampled transitions. **left**: Number of entities changed in the 1-step buffer during training. As expected, the number of transitions with 2 spatial locations changed in the grid increases whereas the ones with no location changed decreases. We also notice a slight increase in the number of transitions with 3 spatial locations changed (corresponding to the agent moving a box!). Training is done in the *fully observed setting*. **right**: Training done in pixel space. Again here, the number of transitions with two spatial locations changed in the grid increases whereas the ones with no location changed decreases. However the number of of transitions with the agent that moves a box did not increase.

We also visualized the transformations learned by both settings. To do so, we manually increased the value of the update gate $f_k$ for a few slots $k$. An example is given in Figure 5 and additional ones are given in section B of the Appendix.

### 4.3 INTRINSIC EXPLORATION STRATEGY

In many environments a uniformly random policy is insufficient to produce action and observation sequences representative enough to be useful for downstream tasks. In this paper we suggest to learn an exploration policy jointly with the model, based on an intrinsic reward that depends on the transition model itself and exploits its entity-centric structure to quantify the diversity of aspects of the environment modified by exploratory behavior. Our model learns to first **select** entities that will be changed and then learns how to **transform** the selected entities. Similar to the empowerment intrinsic objectives (Klyubin et al., 2005; Kumar, 2018), a natural exploration strategy in settings like Sokoban would be to follow trajectories that overall have as many entities being selected as possible. If the agent indeed never pushes a box on target when learning its transition model, it will not be able to transfer its knowledge to a task where it has to push all the boxes on all the targets. We thus suggest to learn a policy that **maximizes the number** of entities selected, as predicted by the current model. We alternate between policy update and model update.

We used a 10-step DQN for the exploration policy and have the DQN and the model share the same 1-step replay buffer. The DQN policy is $\epsilon$-greedy with $\epsilon$ decaying from 1 to 0.3. In order to train the DQN we used the following intrinsic 1-step reward:

$$r(\mathbf{s}_t, a_t) = \sum_k \mathbb{1}_{(f_k^t \geq h)} \qquad (2)$$

with $h$ a chosen threshold for the update gate value. We expect this training strategy to promote trajectories with as many entities that will have their state changed as possible. We thus expect the agent to learn not to get stuck, aim for the boxes, push them etc... In order to validate that intuition, we first conduct experiments in the **fully observed setting**. In this setting we consider the following types of moves:

- *valid_move*: Whenever the agent takes a *move* action in a valid direction, two entities will have their state changed: the initial location of the agent and the next one.
- *valid_push* : Whenever the agent takes a *push* action and a box is available to be pushed in the chosen directions, three entities will have their state changed: the initial location of the agent, the initial location of the box and the next location of the box.

- *blocked_push* : Whenever the agent takes a *push* action when there is no box to push in the chosen direction, nothing happens.
- *blocked_move*: Whenever the agent takes a *move* action in a non-valid direction (against a wall, a box etc...), nothing happens.

With our suggested training strategy we expect the agent to promote trajectories with more transitions of type *valid_move* than *blocked_move* and *blocked_push* and hopefully with the number of *valid_push* transitions increased as well. During training, we thus monitor the true number of entities changed in the transitions stored in the shared 1-step buffer. We also performed the same experiment in the raw input pixels setting and monitored the true number of entities changed in the 1-step buffer during training. Results are reported in Figure 6 and confirm our hypothesis: the agent learns to avoid actions that will result in no changes in the environment (*blocked_push* and *blocked_move*. Details of the hyperparameters are given in Appendix.

## 5 CONCLUSION

We have introduced **SPECTRA**, a novel model to learn a sparse slot-structured transition model. We provided evidence to show that sparsity in the transitions yields models that learns more primitive transformations (rather than configuration-dependent) and thus transfer better to out-of-distribution environments with increased combinatorial complexity. We also demonstrated that the implicit sparsity of the transitions enables an exploration strategy that aims at maximizing the number of entities that be will be modified on the agent's trajectory. In Figure 6 we showed that with this simple exploration strategy the agent leans to avoid actions that will not change the environment (*blocked_move* and *blocked_push*). Preliminary results in pixel space show that SPECTRA biases even a simple perception module towards perceptual groupings that are entity-centric. In Figure 5 we also showed the benefit of jointly training the perception (encoder) module and the transition module. We anticipate that our model could be improved by incorporating a more sophisticated perception module. In the future we aim to use SPECTRA to investigate possible uses in model-based reinforcement learning.

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

## A  ARCHITECTURE AND HYPERPARAMETERS

### A.1  FULLY OBSERVED SETTING

In the fully observed setting the input at time $t$ is a set $\mathbf{o}^t \in \{0,1\}^{N \times 7}$ corresponding to one-hot labels (that can be agent (off and on target), box ( off and on target), wall, target and floor). of each entity in a $7 \times 7$ grid ($N = 49$). We also append their normalized $x - y$ coordinates so that the

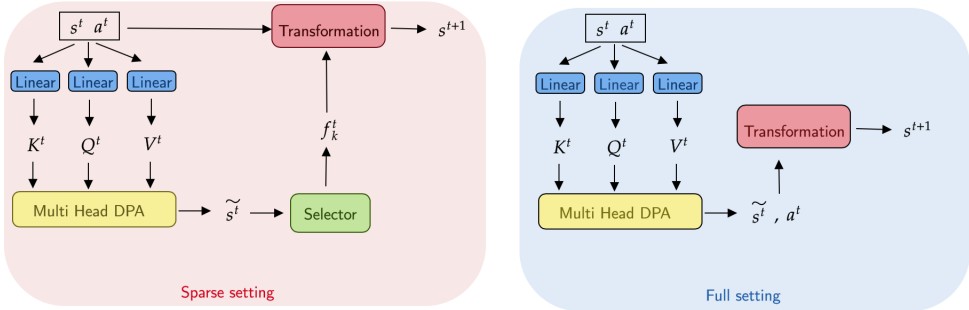

Figure 7: Transition model with and without selection phase.

final input to the transition model is a set $\mathbf{s}^t \in \{0, 1\}^{N \times 9}$. Like detailed previously in Figure 7, the transition model is composed of two modules: the **selection** module and the **transformation** module.

In section 4.1 we also distinguished between the **sparse** and the **full** setting and they are described in 7. In the full setting, there is no more selection bottleneck and the transition module is a simple transformer-like architecture.

**Selection module.** The selection module is a transformer-like architecture. It takes as input at time step $t$ the concatenation $\mathbf{e}^t = [\mathbf{s}^t, a^t]$ of the set $\mathbf{s}^t$ and the action $a_t$. The selection module is then composed of 2 attention heads where is head is stack of 3 attention blocks (Vaswani et al., 2017; Zambaldi et al., 2018). The 3 blocks are 1-layer MLP that output key, query and value vectors of channels size 32, 64, 64 respectively. The first two blocks are followed by $RELU$ non linearities and the last one doesn't have any. The output of the attention phase is thus the concatenation of values obtained from the 2 attentions heads $\tilde{\mathbf{s}}^t \in \mathbb{R}^{N \times 112}$. To obtain the selection binary selection variables we then simply apply slot-wise a single layer MLP to the concatenation $\tilde{\mathbf{e}}^t = [\tilde{\mathbf{s}}^t, a^t]$ followed by a logSoftmax non-linearity in order to compute the log-probabilities of each entity to be modified by the action taken. The output of the selection module is thus a set of log-probabilities $\mathbf{l}^t \in \mathbb{R}^{N \times 2}$.

**Transformation module.** The transformation module is a simple shared 2-layers MLP that is applied slot-wise to the the concatenation $\mathbf{e}^t = [\mathbf{s}^t, a^t]$ of the input set $\mathbf{s}^t \in \{0, 1\}^{N \times 9}$ and the action taken. It outputs channels of sizes 16, 7 respectively. The first layer is followed by a $RELU$ non-linearity and the last one by a logSoftmax non-linearity in order to compute the log-probabilities of the label of each predicted entity.

**Full setting.** In the full setting, we don't have a selection bottleneck anymore. The transformation module is thus directly applied to the output of the attention phase $\tilde{\mathbf{e}}^t = [\tilde{\mathbf{s}}^t, a^t]$. It consits this time of a simple shared 3-layers MLP that is applied slot-wise and outputs channels of sizes 64, 32, 7 respectively. The first two layers are followed by a $RELU$ non-linearity and the last one by a logSoftmax non-linearity .

## A.2 LATENT SETTING

In the latent setting the input at time $t$ is a raw pixels (RGB) image $\mathbf{o}^t \in \mathbb{R}^{112 \times 112 \times 3}$. In the latent setting, the transition model is composed of a **perception** module, a **selection** module and a **transformation** module.

**Perception module.** When dealing with unstructured input we first need a way to extract entities latent representations. For this work we used a very simple and naive perception module, with an

encoder similar to what is done by Zambaldi et al. (2018); Santoro et al. (2017). Like detailed in Figure 1, we use a CNN to parse pixel inputs into $k$ feature maps of size $n \times n$, where $k$ is the number of output channels of the CNN. We choose arbitrarily $n = 4$ and didn't perform any hyperparameter search for the CNN architecture. We then concatenate x and y coordinates to each k-dimensional pixel feature-vector to indicate the pixels position in the map. We treat the resulting pixel-feature vectors as the set of entities $\mathbf{s}^t \in \mathbb{R}^{N \times k}$ where here $N = n^2 = 16$. We denote as $\mathbf{s}^t_{coord} \in \mathbb{R}^{N \times k+2}$ the entities set to which we have appended the x-y position in the map.

As our loss is a pixel loss we also need a decoder that decodes each entity $s^t_{k,coord}$ of the set $\mathbf{s}^t$ back to its corresponding mean $\mu^t_k$ and mask $m^t_k$. The CNN of the encoder outputs channels of size (16, 32, 32, 32, 32). All layers (except the last one) are followed by $RELU$ non-linearities. Kernel sizes are (3, 3, 4, 3) and strides (2, 2, 2, 2, 1). The decoder is composed of a 2-layers MLP followed by a stack of transposed convolutions. The MLP outputs channels of sizes $(7 \times 34, 7 \times 7 \times 34)$ with a $RELU$ non-linearity between the 2 layers. The output is then resized to $7 \times 7 \times 34$ map that will be fed to the convolution part. For the convolution part, it outputs maps of channel sizes (4, 4, 4, 4, 4) with $RELU$ non-linearities between each layer. The kernel sizes are (3, 3, 5, 4).

**Selection and Tranformation modules.** The selection and transformation module are very similar to the fully observed setting, except that they operate on the latent space, so we do not apply LogSofmax non-linearities for the transformation part. The input of the selection module is $\mathbf{s}^t_{coord}$ and the input to the transformation module is $\mathbf{s}^t$. The **selection** module is composed of 2 attention heads where is head is stack of 3 attention blocks (Vaswani et al., 2017; Zambaldi et al., 2018). The 3 blocks are 1-layer MLP that output key, query and value vectors of channels size 34, 16, 16 respectively. The first two blocks are followed by $RELU$ non linearities and the last one doesn't have any. The output of the attention phase is thus the concatenation of values obtained from the 2 attentions heads $\tilde{\mathbf{s}}^t \in \mathbb{R}^{N \times 32}$. To obtain the selection binary selection variables we then simply apply slot-wise a 3-layers MLP of channels sizes 16, 32, 32 respectively to the concatenation $\tilde{\mathbf{e}}^t = [\tilde{\mathbf{s}}^t, a^t]$ followed by a Softmax non-linearity in order to compute the probabilities of each entity to be modified by the action taken. The output of the selection module is thus a set of probabilities $\mathbf{p}^t \in \mathbb{R}^{N \times 2}$. The **transformation** module is a simple 2-layers MLP of channels sizes 32,32 respectively with a $RELU$ non-linearity between the two layers.

## B    ADDITIONAL VISUALISATIONS

In this section we reported additional visualizations similar to Figure 3 and 5 where we monitor:

- Differences in slot-wise masked decodings of the perception module when it is trained jointly and separately from the sparse transitions.
- Differences in the slot-wise transformations earned by the transition model when it is trained separately and jointly with the perception module.

We notice that joint training enables to learn slot-structured representation that are **entity-centric** and thus enable to learn better transition models. The transformations learned are especially visually more *interpretable*.

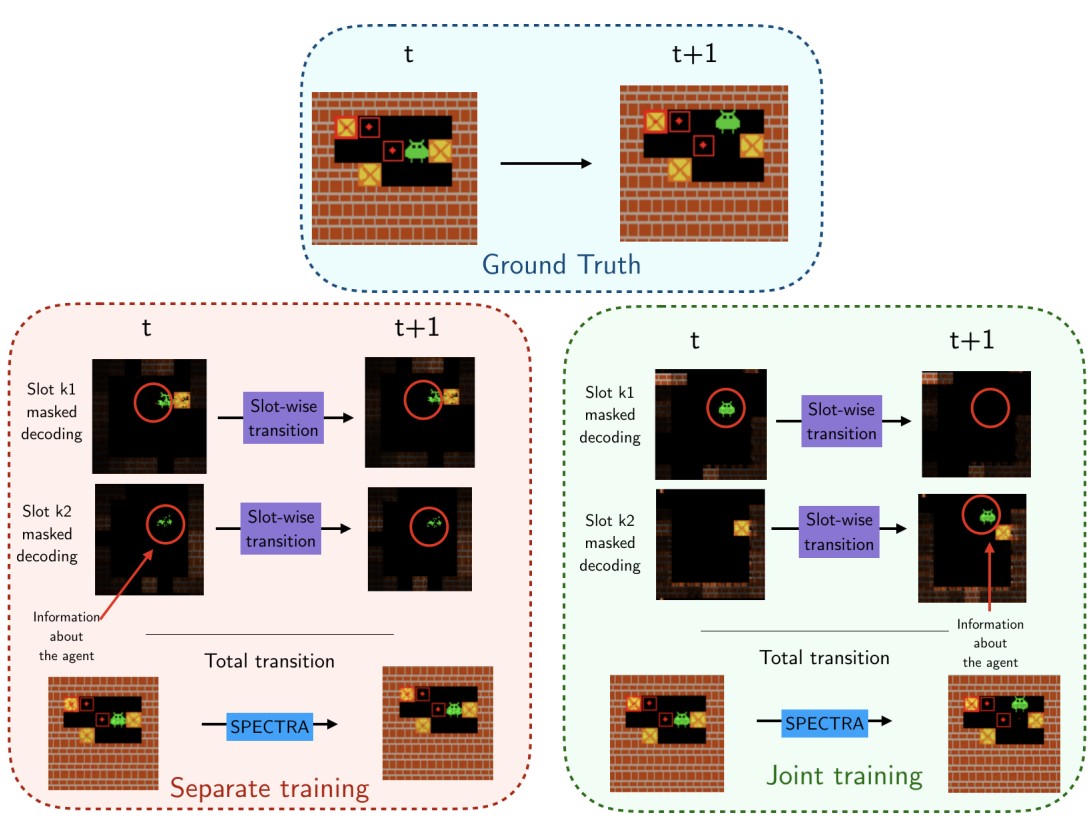

Figure 8: Additional visualisations of masked decodings from joint and separate training settings.

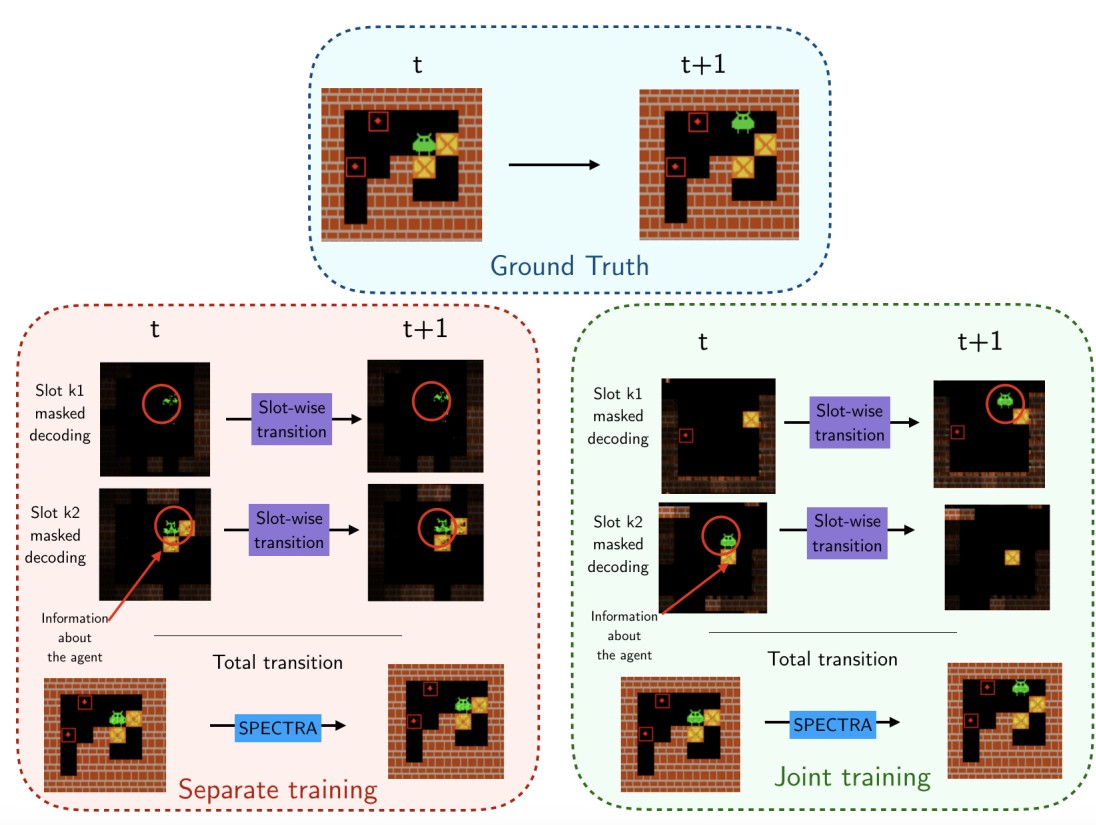

Figure 9: Additional visualisations of masked decodings from joint and separate training settings.

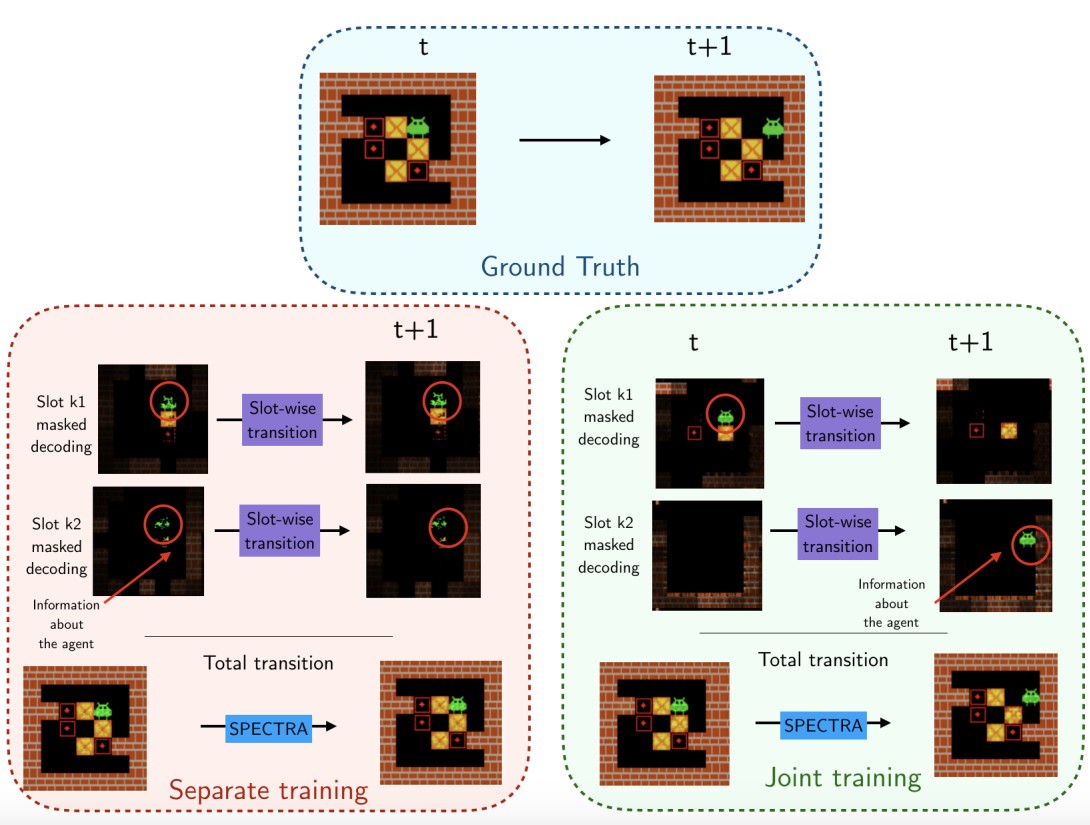

Figure 10: Additional visualisations of masked decodings from joint and separate training settings.

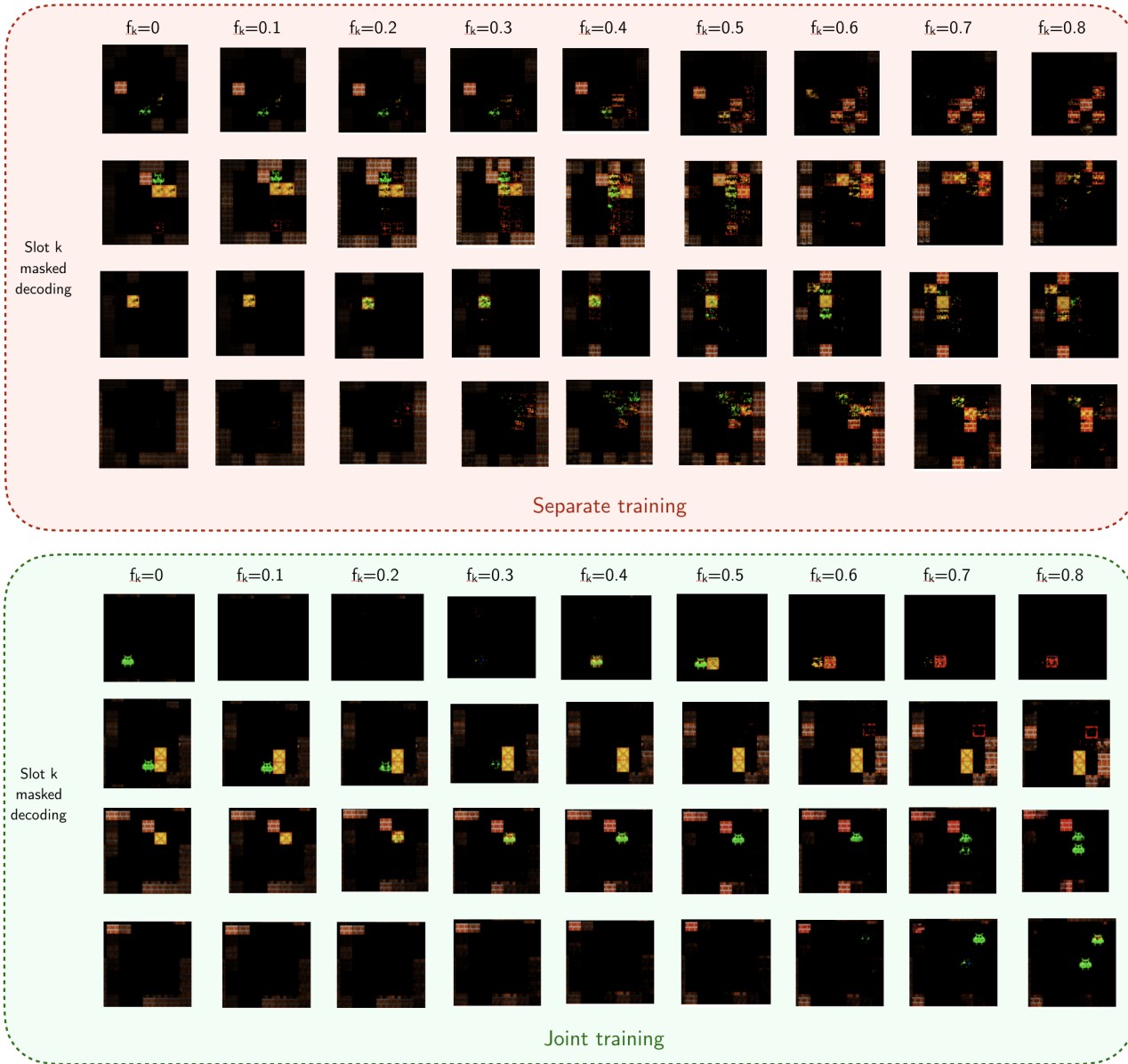

Figure 11: Additional visualisations of the transformations learned when the perception module is trained jointly and separately from the transitions. Joint training yield more visually interpretable and localized transformations of the slots.

