# OpenReview forum: "SPECTRA: Sparse Entity-centric Transitions"
_ICLR.cc/2020/Conference — Reject_

### Official Review · AnonReviewer1 · 2019-10-23
**Official Blind Review #1**

**Rating:** 3

**Review:**

This paper proposes a model that learns to disentangle visual scene into objects (slots), and simultaneously learns a dynamics model to capture how these objects interact with each other.  The authors demonstrated that the proposed model has the potentials to discover objects without supervision, and also enables an exploration strategy that maximizes the number of objects changed in the agents' trajectories.

This paper is for sure studying an important problem. The approach presented in the manuscript makes intuitive sense. The experimental results are reasonable but can be strengthened. However, I was shocked that the authors seem to be unaware of the abundant related work in this area (see below). I'd like to see the authors' responses regarding the missing related work. As of now, my recommendation is a clear reject.

As the authors mentioned, objects play a key role in human perception, and the problem of discovering objects from visual input is for sure an important problem. I commend the authors for pursuing research in this direction.

The experimental evaluations are however a little limited: it's restricted to games, where the visual appearance of objects is almost identical. In those cases, it'd be hard to access how the model may generalize to real-world data, where object appearances and texture can be complex. There are also no comparisons with published, SOTA methods.

The major problem of this manuscript, to me, is its ignorance of related work and, therefore, overclaiming at a few places. Most importantly, I suggest the authors cite, discuss, and ideally compare with many related works from Josh Tenenbaum's group and Sergey Levine's group. A few papers listed below, in particular [A] and [B], also formulated the problem in a similar fashion, where they decompose the scene into object-centric representations and learn the interactions among objects. [B] and [C] also explored how the model can be used in an RL setting. Further, [D] studied learning an object-oriented dynamics predictor in a similar context as presented in this paper.

The sparse effects have also been explored by Xia et al. [E]. Similarly, the work has built upon object-centric representations. I, therefore, wonder how the proposed method compares with all those published papers, both at the conceptual level and at the experimental level.

[A] Wu et al. Learning to See Physics via Visual De-animation. NeurIPS 2017
[B] Janner et al. Reasoning About Physical Interactions with Object-Oriented Prediction and Planning. ICLR 2019
[C] Co-Reyes et al. Discovering, Predicting, and Planning with Objects. ICML 2019 Workshop, https://sites.google.com/view/dpppo/
[D] Zhu et al. Object-Oriented Dynamics Predictor. NeurIPS 2018
[E] Xia et al. Learning sparse relational transition models. ICLR 2019


**Experience Assessment:**

I have published in this field for several years.

**Review Assessment: Checking Correctness Of Derivations And Theory:**

I carefully checked the derivations and theory.

**Review Assessment: Checking Correctness Of Experiments:**

I assessed the sensibility of the experiments.

**Review Assessment: Thoroughness In Paper Reading:**

I read the paper thoroughly.

---

### Official Review · AnonReviewer3 · 2019-10-23
**Official Blind Review #3**

**Rating:** 3

**Review:**

This paper introduces a model that learns a slot-based representation, along with a transition model to predict the evolution of these representations in a sparse fashion, all in a fully unsupervised way. This is done by leveraging a self-attention mechanism to decide which slots should be updated in a given transition, leaving the others untouched. The model learns to encode in a slot-wise and is trained on single step transitions.

This work tackles an important problem, and is very well motivated and presented in a very clear fashion. It reuses some known ideas and components, but combines them in a nice way. I especially liked the use of attention to select what to update, which is a good prior to have.

However, the results presented unfortunately seem to fall a bit short in this current version, and some decisions might have had too much of an effect on some of these shortcomings. Given some improvements, this work might become quite promising, but for the time being I am leaning against publication.

1.	Most modeling decisions are clear and well-motivated, however the choice to make the transition model f_trans always be applied only “slot-wise” might be too restrictive. Indeed, for a given action, this means that 2 slots have to independently learn the effect of that action (e.g. as shown in the example in Figure 2. right), and that some interactions are ~impossible to learn (e.g. in Sokoban, pushing a box requires knowing about the location of both the agent and the box). This could have been alleviated if the transition had access to “interactions outcomes” (e.g. if using a GraphNet, or in your model, if \tilde{s} was provided to the transition function f_theta). Other works (including Zambaldi et al 2018, which is cited several times), handle this appropriately. Did you try to provide \tilde{s} to the transformation operator (e.g. in Figure 7 left)?
2.	Adding a direct comparison to pure non-slotted versions of the model/baselines would have been quite useful, as currently it is unclear why certain things are failing.
3.	Similarly, finding what was the output of the CNN encoder for pixel inputs was a bit too difficult. It is explained in the Appendix that one maps into 4x4 feature maps, but that might be too large for the current environments? Indeed for Sokoban, this means that any grid is partially supported by several “slots”, which may hurt the results more than they should (especially combined with the slot-wise transition constraints expressed above).
4.	The early state of the current results are quite visible in all examples of the “Separate training” model predictions (Figure 3, 5, 8, 9, 10 and 11). None of these actually show this model performing a “correct” prediction for t+1? They only predict no changes, or nonsensical interpolations… This is not sufficient to try to make an argument about the “joint training” helping, and most discussions about “what information they contain” is strenuous at best.
5.	The paper keeps mentioning that it “implicitly imposes transitions to be sparse”, however it is never explained how that would come about? I understand that the softmax in the self-attention may tend to become “peaky” and hence only affect a few slots (and the results do seem to confirm this observation), but I was expecting an explicit loss to enforce this fact. The current emphasis seems a bit ill-funded, so I would present more evidence to it or downplay it.
6.	Some of the results shown seem hard to interpret or provide only weak evidence for the proposed model:
	a.	Figure 2. Left does seem to indicate a benefit in using the self-attention module, but it is hard to know how much of an effect the gap between the orange and red curves actually imply. This figure is overall a bit too small to interpret, and it might be better to split the 2 conditions into sub-plots. The names of the curves in the legend do not correspond to anything described in the text/caption (but I could understand them…). I was expecting more discussion of the results in Figure 2, for example at the end of Section 4.1.
	b.	As explained above, Figure 3 only shows that the Joint Training can perform a 1-step prediction, which is ok but is the bare minimum. Does it handle multi-step rollouts?
	c.	Figure 4 does not seem to provide any significant results or insights. I would interpret them as showing no significant difference between the curves, and they are too small to extract any information out of them. I would remove this figure fully.
	d.	Figure 5 is unclear about what f_k=0 really is, and once again just shows that Separate training does nothing. I expected it to be exactly reconstructing s_t (given that the others are trying to predict s_t+1)? But this is only the case for the joint training, and without knowing what x_t actually was, it is hard to trust. The fact that f_k changes what it does as it is being increased makes the whole point hard to interpret.
7.	Figure 6 was quite interesting, and I feel like this could be pushed forward in a quite interesting manner. The choice of “maximizing the number of entities selected” was fair as a first try, but I could imagine it failing to generalize to more complex environments, or to be easily exploited if one ever decides to pass gradients back into the representation from the policy. It was unfortunate that the legends do not correspond to anything expressed in the main text or caption (e.g. why do you not reuse the “valid_move”, …, ”blocked_push” names that you thoroughly introduce?). What is “random_XX”? Could you comment on why the curves seem to have a large increase in variance along the 80000-100000 updates region?


Details:
8.	Figure 2 (right) was good to explain how the model worked, but I would actually change its location and try to move it into Figure 1. Similarly, Figure 7 in the Appendix seemed rather necessary to understand the model, and belongs in the main text in my opinion.
9.	When presenting the self-attention block, it would be good to directly state that these are MLPs receiving [s_t, a_t] (as done in the Appendix).
10.	Is sigma^2 in the decoder fixed? To which value?
11.	When presenting the “sparse” and “full” settings, having access to Figure 7 and the rest of the Appendix might be beneficial, it would be good to point forward to it.



**Experience Assessment:**

I have published in this field for several years.

**Review Assessment: Checking Correctness Of Derivations And Theory:**

I carefully checked the derivations and theory.

**Review Assessment: Checking Correctness Of Experiments:**

I carefully checked the experiments.

**Review Assessment: Thoroughness In Paper Reading:**

I read the paper thoroughly.

---

### Official Review · AnonReviewer2 · 2019-10-26
**Official Blind Review #2**

**Rating:** 3

**Review:**

This paper proposes to use a ‘slot-based’ (factored) representation of a ‘scene’ s.t. a forward model learned over some observed transitions only requires sparse updates to the current representation. The results show that jointly learning the forward model and the scene representation encourages meaningful ‘entities’ to emerge in each slot. Additionally, the paper argues that this representation allows for better generalization and can also guide exploration by rewarding actions that change multiple entities

I really like the overall idea i.e. jointly learning the scene representation and the transition model, while enforcing sparse transitions. The experiments in Sec 4.2 and the visual results in Fig 3 do clearly highlight the benefits of joint learning, and show that the emergent representations are more meaningful compared to learning representations independently.

However, while the overall approach is intuitive and seems to yield desirable results, I have concerns regarding the experiments, comparisons to prior work, and the exact contributions of this work. Specifically:

1) It is unclear what this paper is claiming to improve over prior work: is the goal to a) learn a good forward model, or b) show that emergent entities allow better downstream tasks. If it is ‘a’ , then there are several other ways of learning good forward models e.g. convolution flow-based [1], and simply showing comparisons to a naive baseline is not sufficient. Similarly, the only downstream task examined is exploration, and again the only comparison is against a variant of the method. Therefore, while the results regarding the emergent representation are good compared to a variant without joint training, they are not shown to be useful in context of any task when compared to the approaches in literature.

2) Despite the motivation in the introduction regarding applications to RL, the paper essentially learns a specific form of factored forward model. There have been several prior works which also pursue a similar approach (though mainly in context of video prediction) e.g. [2], and I don’t think this paper’s approach is novel in context of these. In any case, some comparisons should be made to these form of models which also show emergent entities with a graph-structured transition model.

3) While the experiments in Sec 4.2 clearly demonstrate the benefits of the approach, the ones in Sec 4.1 and 4.3 are less convincing:
3a) Sec 4.1 shows that the slot based transition model generalizes better, but this is only in comparison to a naive fully-connected baseline. I feel any prediction model with some structure e.g. graph-based forward model, convolutional forward model etc. would also similarly generalize.
3b) Sec 4.2 argues that this slot-based representation can help exploration, but this is in fact a chicken-and-egg problem, as one needs to have collected interesting transition samples for a good representation to emerge. In fact, the results shown in Fig 6b) show that actions affecting 3 blocks were not explored, perhaps because the random transitions did not sample these to begin with.

Overall, the approach proposed is desirable, but there are closely related alternatives that exist in literature, and there need to be more concrete comparisons against these for either prediction quality or success in downstream task.

---
References:
[1] Self-Supervised Visual Planning with Temporal Skip Connections, Ebert et. al, CORL 17
[2] Learning to decompose and disentangle representations for video prediction, Hsieh et. al., NeurIPS 18

**Experience Assessment:**

I have read many papers in this area.

**Review Assessment: Checking Correctness Of Derivations And Theory:**

I assessed the sensibility of the derivations and theory.

**Review Assessment: Checking Correctness Of Experiments:**

I carefully checked the experiments.

**Review Assessment: Thoroughness In Paper Reading:**

I read the paper thoroughly.

---

### Author Response · Authors · 2019-11-15
**Thanks**

We would like to thank the reviewers for all their useful comments. We especially agree with Reviewer 2 regarding the prior work we claim to improve upon (representation for some RL downstream tasks vs accuracy of the forward model learned). We will come with a stronger body of experiments for a future submission. We also thank Reviewer 3 for his very detailed comments on all the Figures in the draft and Reviewer 1 for pointing out some missing related work. However regarding those, we were not aware of [C] until they released the arXiv version a few days ago, [B] uses ground truth objects segmentations to learn object-centric representations, [D] doesn’t seem to have a bottleneck for each object and [E] seems to use high-level representations of the blocks and thus doesn't focus on representation learning.

---

### Decision · Program_Chairs · 2019-12-19

**Decision:**

Reject

**Comment:**

This paper introduces a model that learns a slot-based representation and its transition model to predict the representation changes over time. While all the reviewers agree that this paper is focusing on an important problem, they expressed multiple concerns regarding the novelty of the approach as well as lacking experiments. It certainly is missing multiple important relevant works, thereby overclaiming at a few places. The authors provided a short general response to compare their approach with some of the previous works and conduct stronger experiments for a future submission. We believe this paper is not at the stage to be published at this point.